# Effectiveness and Nephrotoxicity of Long-Term Tacrolimus Administration in Patients with Ulcerative Colitis

**DOI:** 10.3390/jcm9061771

**Published:** 2020-06-07

**Authors:** Keiichi Haga, Tomoyoshi Shibuya, Kei Nomura, Koki Okahara, Osamu Nomura, Dai Ishikawa, Naoto Sakamoto, Taro Osada, Akihito Nagahara

**Affiliations:** Department of Gastroenterology, Juntendo University School of Medicine, 2-2-1 Hongo, Bunkyoku, Tokyo 113-8421, Japan; khaga@juntendo.ac.jp (K.H.); ke-nomura@juntendo.ac.jp (K.N.); k-okahara@juntendo.ac.jp (K.O.); onomura@juntendo.ac.jp (O.N.); dai@juntendo.ac.jp (D.I.); sakamoto@juntendo.ac.jp (N.S.); otaro@juntendo.ac.jp (T.O.); nagahara@juntendo.ac.jp (A.N.)

**Keywords:** inflammatory bowel disease, ulcerative colitis, tacrolimus, renal function

## Abstract

Background: Tacrolimus (TAC) is used for the management of ulcerative colitis (UC). However, there are few reports on the effectiveness of its long-term administration. TAC is also known to cause renal toxicity. The aim of this study was to evaluate long-term effectiveness and monitor changes in renal function during prolonged TAC use in patients with UC. Methods: Medical records of 50 UC patients treated with TAC were retrospectively reviewed. Clinical outcomes were assessed at 6, 12, 24, and 36 months after initiating TAC. We also monitored chronological changes in renal function. Results: Thirty-nine patients were treated with TAC for more than 3 months. Relapse-free survival among these patients at 6, 12, 24, and 36 months was 82%, 69%, 41%, and 23%, respectively. On the other hand, renal function was reduced in 35.9% of patients. We found that irreversible renal dysfunction was more likely to occur in cases in which the estimated glomerular-filtration rate (eGFR) was reduced by more than 30%. Conclusion: This study demonstrated the potential use of TAC as an effective option in the long-term medical management of UC, although it tended to increase the risk of nephrotoxicity. There is a need for the careful monitoring of renal function during TAC administration.

## 1. Introduction

Ulcerative colitis (UC) is a chronic relapsing and remitting inflammatory disorder of the large intestine. In the treatment of UC, corticosteroids are still important basic therapies [1]. However, it was reported that 20% of patients with UC become steroid-dependent after initiating corticosteroid treatment [2,3]. Various treatments have become available in recent years, and calcineurin inhibitors are important in the treatment of steroid-resistant or -dependent cases as well as antitumor necrosis factor-α (TNF-α) antibody preparations [4,5,6]. 

Tacrolimus (TAC) is a calcineurin inhibitor isolated from soil bacterium *Streptomyces tsukubaensis* [7]. TAC has immunosuppressive effects by suppressing cellular functions such as cytokine production through T-cell activation, and it is well known that T-cell dysfunction plays a crucial role in the pathogenesis of inflammatory bowel disease (IBD) [8,9]. TAC possesses potent immunosuppressive properties, and it has been commonly used as an immunosuppressant to prevent kidney- or liver-transplant rejection [10]. Effectiveness against UC has been suggested since the late 1990s [11]. In Japan, TAC was approved for clinical use in UC patients in 2009. TAC has been demonstrated to have remarkable short-term therapeutic efficacy [8,12,13,14]. The administration period is up to 3 months for UC, but reductions in dosages have resulted in worsening in many cases, and in some cases, long-term administration is unavoidable. Several retrospective studies showed the long-term effectiveness of TAC therapy [8,15,16,17,18,19,20]. However, evidence of the efficacy and safety of long-term administration is still inadequate. Despite well-documented safety records, patients have been known to develop side effects, namely, infection, hypertension, nephrotoxicity and various neuropsychiatric problems [21]. Nephrotoxicity is one of the important side effects of TAC administration, along with infection [22]. Therefore, regular safety monitoring, including the measurement of plasma trough TAC levels, is required. However, few reports have focused on renal damage caused by TAC in UC patients. The aim of this study was to evaluate the long-term effectiveness of TAC, and monitor changes in renal function during its prolonged use in patients with UC.

## 2. Materials and Methods

### 2.1. Study Design

Data were compiled from 50 moderate-to-severe active UC patients treated with TAC in Juntendo University between April 2010 and December 2018. Their medical records were reviewed retrospectively for diagnosis, clinical course, treatment, and renal function. Patient data were registered into an electronic database after a de-identification process. The protocol for this retrospective investigation was reviewed and approved by the Juntendo University Hospital Ethics Committee (IRB no. 20-014). This study adhered to the principles of the Declaration of Helsinki.

### 2.2. Patients

UC diagnosis was based on established standardized criteria by prior clinical assessment, radiology, endoscopy, and histology. Patients with prior steroid treatment were classified as having steroid-resistant or -dependent UC. Steroid-resistant UC is defined as an active disease despite a prednisolone dose of up to 0.75 mg/kg/d over a period of 4 weeks. Steroid-dependent UC is defined as a lack of ability to reduce steroids below the equivalent of 10 mg/d prednisolone within 3 months of starting steroids, without recurrence of disease activity or relapse within 3 months of stopping steroids [23]. 

### 2.3. Treatment

TAC was administered in its oral formulation. In all cases, TAC was initiated at a dosage of 0.05–0.1 mg/kg/day, aiming for a target trough level of 10–15 ng/mL for the first 2 weeks to induce remission. After inducing clinical remission, TAC whole-blood levels were maintained at a range of 5–10 ng/mL [5]. 

### 2.4. Definition of Response

Clinical outcomes were assessed at 3, 6, 12, 24, and 36 months after initiating TAC. Clinical disease activity was determined using the Lichtiger clinical-activity index (CAI) [24]. The Lichtiger CAI is composed of the following items: number of daily bowel movements, abdominal pain and tenderness, use of antidiarrheics, blood in stools, general well-being, fecal incontinence, and nocturnal diarrhea. A higher score indicates a more severe disease (score range 0–21). CAI ≥ 10, 7–9, and ≤6 were defined as severe, moderate, and mild, respectively [24,25,26]. Endoscopic severity was determined using the Mayo endoscopic-subscore (Mayo ES) classification (0, normal or inactive disease; 1, mild disease with erythema, decreased vascular pattern, mild friability; 2, moderate disease with marked erythema, absence of vascular patterns, friability, erosions; 3, severe disease with spontaneous bleeding, ulceration) [27] and the ulcerative-colitis endoscopic index of severity (UCEIS) [28]. The UCEIS is scored using the three following items: vascular pattern, bleeding, and erosions and ulcers. A score of 0 to 3 is given for each item, with a higher score indicating more severe endoscopic activity (score range 0 to 8). Mucosal healing was defined as a Mayo ES of 0 or 1 [29]. Clinical response was defined as a decrease in the CAI of 3 or more points, and clinical remission was defined as a score of 3 or fewer points. Maintenance of efficacy was defined as no exacerbation of CAI and no need for the increased intensification of treatments [30]. 

### 2.5. Renal Function

We monitored chronological changes in renal function by following the estimated glomerular-filtration rate (eGFR) and the serum creatinine level during TAC administration, and compared changes in eGFR and serum creatinine after TAC discontinuation. Renal failure was defined as a 30% decrease in the eGFR relative to baseline. Associations of age, sex, plasma trough TAC level, and dose with renal function were examined.

### 2.6. Statistical Analysis

All data were analyzed using GraphPad Prism (version 6, GraphPad, La Jolla, CA, USA). Differences between groups were analyzed using Mann–Whitney’s U test. Relapse-free survival was assessed using the Kaplan–Meier method. Significance was defined at *p* < 0.05. 

## 3. Results

### 3.1. Patient Characteristics

Characteristics of the 50 patients included in this study are shown in Table 1. There were 29 males and 21 females, with a median age of 37.5 years (range 18–68 years). The median duration of the disease was 6 years (range 1–33 years). Nineteen and 31 patients were classified as having left-sided or extensive colitis, respectively. The median Lichtiger CAI at baseline was 12 (range 8–16). Forty-five patients were treated with aminosalicylate prior to TAC treatment, and 4 patients switched medication from infliximab to TAC. Fourteen patients (28%) were steroid-refractory, 34 (68%) were steroid-dependent, and 2 (4%) were steroid-naïve. The median duration of TAC therapy was 13.5 months (range 1–64 months) (Table 1). No patient had a history of renal disease. Five patients had family histories of inflammatory bowel disease, and one patient of renal disease.

### 3.2. Treatment Efficacy

Among patients who achieved a clinical response (CR) at 3 months after initiating TAC, 39 (78%) maintained the CR while continuing TAC. These included patients who did not have other treatment options or whose condition worsened upon reducing the dose of TAC. Drug withdrawal was observed in 11 patients at 3 months; 8 nonresponders, 1 patient by self-determination and 2 patients due to adverse events such as acute nephrotoxicity and nausea (Figure 1). Relapse-free survival among these patients at 6, 12, 24, and 36 months was 82%, 69%, 41%, and 23%, respectively (Figure 2). Three patients needed total colectomy during the observation period (Figure 1). Median Lichtiger CAIs at 3, 6, 12, 24, and 36 months were 2 (range 0–11), 3 (range 0–6), 3 (range 0–8), 3 (range 0–11), and 3 (range 0–8), respectively. We next assessed endoscopic improvement in cases that underwent colonoscopy. Median UCEIS was 5 (range 3–8), and median Mayo ES was 3 (range 2–3) at baseline. Mucosal healing (Mayo ES 0 or 1) was achieved by 24 of 30 patients (80%) 6 months after the introduction of TAC. Among patients achieving a CR at 12 months, mucosal healing was maintained in 19 of 27 patients, and in 12 of 16 patients at 24 months (Figure 3).

### 3.3. Renal Function

The average eGFR and serum creatinine at baseline were 108.4 mL/min/1.73 m^2^ and 0.61 mg/dL, respectively. Renal function was reduced in 35.9% (14/39) of patients showing a 30% decrease in eGFR relative to the baseline during the observation period. The average oral dose after 3 months of administration was 5 mg/day. The mean plasma trough TAC level was controlled at 5–10 ng/mL and remained flat over time (Figure 4). Although nephropathy is usually dependent on the dosage of TAC, eGFR gradually decreased in cases treated with long-term TAC administration, even though plasma trough TAC levels remained flat (Figure 4 and Figure 5a). Of patients with a 30% decrease in eGFR, three patients had a drop in eGFR of more than 60% (Figure 5b). Changes in renal function were analyzed in 23 cases treated with TAC for more than 6 months and whose course could be followed for more than 1 year after the discontinuation of TAC treatment. Renal function tended to improve after discontinuation of TAC in 19 cases. However, eGFR remained reduced more than 30% compared to values before the administration of TAC even after TAC had been discontinued for 1 year in 4 cases (Figure 6a). Moreover, although renal function was within normal range, eGFR remained reduced after the discontinuation of TAC in cases with eGFR decreases greater than 30% (*p* < 0.05) (Figure 6b).

## 4. Discussion

The goals of UC management are to maintain clinical remission and mucosal healing while avoiding hospitalization and surgery. TAC is used for the management of refractory UC and has excellent short-term efficacy [8,12,13,14]. Although maintenance therapy with thiopurine is usually recommended after induction of remission with TAC, several studies evaluated the long-term efficacy of TAC for maintaining remission [8,15,16,17,18,19].

Yamamoto et al. reported the results of retrospective analysis of UC patients who maintained remission with either TAC or thiopurine. There was no significant difference in the therapeutic effect between the two agents. However, the relapse-free survival rate was significantly lower in the thiopurine-refractory TAC group than in the thiopurine group [18]. This report is interesting because it shows that TAC may be an option for the maintenance of remission in patients who are intolerant of immunomodulators. Additionally, some other retrospective studies reported the usefulness of TAC as a maintenance therapy [8,17,20,31,32]. Endo et al. reported the long-term therapeutic efficacy of TAC, and compared therapeutic efficacies between patients treated with infliximab (IFX) and TAC. Though IFX and TAC showed similar short-term efficacy, the IFX-based strategy had better outcomes with regard to long-term efficacy for the treatment of steroid-refractory UC [31]. According to studies reporting long-term prognosis with TAC treatment, eventfree survival rates were 57–82%, 37–56%, and 38–50% at 6, 12, and 24 months, respectively [17,20,31,32]. In our study, relapse-free survival among patients treated with TAC at 6, 12, and 24 months was 82%, 69%, and 41%, respectively. Although patients’ backgrounds differed between our studies and those of the others, our treatment outcomes were consistent with those in previous reports. Furthermore, in this study, patients who maintained clinical remission had a high rate of mucosal healing. These data suggest that TAC may have potential as a maintenance therapy.

Although TAC is useful for intractable UC, its prolonged use appeared to increase the risk of nephrotoxicity. However, there are few reports of TAC-induced renal damage in UC. A cohort study of patients who had undergone transplantation of organs other than kidneys revealed that 16.5% of chronic renal dysfunction occurred during a 36-month observation period. Surprisingly, 28.9% of these patients were reported to have progressed to end-stage renal failure [33]. According to the survey of specific uses of TAC for UC, the incidence of renal dysfunction was seen in 8.4% of the patients. Among these cases, 16.1% were reported to be severe. 

TAC can cause acute or chronic nephrotoxicity, particularly when used in high doses or over a long period. Acute nephrotoxicity due to calcineurin inhibitors is mainly observed early in administration and caused by the vasoconstitution of imported arterioles. Vasoconstitution is thought to be caused by an imbalance of factors such as endothelin, thromboxane, activation of the renin–angiotensin system, or vasodilators such as prostaglandin E2, prostacyclin, and nitric oxide. These changes depend on the dose of the calcineurin inhibitor and are reversible [34]. Therefore, reducing dose and lowering trough concentration are usually needed when nephrotoxicity is observed during TAC administration. In our study, mean plasma trough TAC level was controlled at 5–10 ng/mL, and the TAC dose or trough level remained flat over time (Figure 4). However, eGFR gradually decreased during long-term TAC administration (Figure 5a). These data showed that TAC troughs and doses do not always reflect renal function during prolonged TAC use. 

Chronic nephropathy is characterized by renal tubular interstitial fibrosis and can lead to irreversible nephrotoxicity, although the mechanism is not as clear as that for acute nephropathy [35]. eGFR is known to be more sensitive than serum creatinine levels to observed changes in renal function during prolonged TAC dosing. However, caution is needed, as eGFR could be impaired, but could still remain within a normal range in some cases. In our study, renal function was reduced in 35.9% (14/39) of patients with more than 30% decrease in eGFR relative to the baseline. However, eGFR was still observed to be within the normal range (Figure 5a). Remarkably, eGFR remained reduced even after the discontinuation of TAC in cases when eGFR was decreased by more than 30%, while eGFR improved in cases where the decrease in eGFR remained within 30%. 

Our study demonstrated that the discontinuation of TAC treatment should be considered before eGFR is decreased by more than 30% during TAC administration. While TAC has the potential for maintenance therapy, we must not forget the risk of nephrotoxicity. Since TAC troughs and doses do not always reflect renal function, eGFR is an important predictor of kidney reserve, and it is necessary to regularly evaluate eGFR. Detection of nephrotoxicity may enable us to appropriately adjust treatment. Limitations of our study included the sample size, retrospective design, single-center experience, and the heterogeneity of previous and current treatments among patients. Urine tests were also not examined in this study. Further studies are needed to elucidate the cause of irreversible renal damage due to the long-term administration of TAC.

## 5. Conclusions

In conclusion, long-term administration of TAC appeared to prevent the relapse of UC. There are still no clinical trials that prospectively examined the effectiveness of the long-term administration of TAC, and it is necessary to accumulate evidence on the usefulness and safety of TAC dosing for the maintenance of remission. However, this study demonstrated the potential use of TAC as an effective option in the long-term medical management of patients with UC. On the other hand, prolonged use of TAC tended to increase the risk of nephrotoxicity. In most cases, renal function improved upon discontinuation or reduction of the dose, but in some patients, there was a significant decrease. There is a need for careful monitoring of renal function during TAC dosing. 

## Figures and Tables

**Figure 1 jcm-09-01771-f001:**
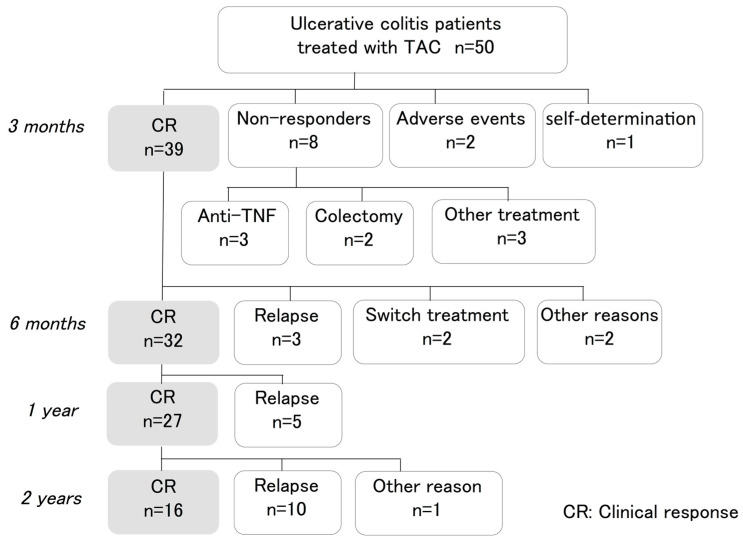
Flowchart of treatment outcomes. Clinical outcomes were assessed at 3, 6, 12, 24, and 36 months after initiating tacrolimus (TAC).

**Figure 2 jcm-09-01771-f002:**
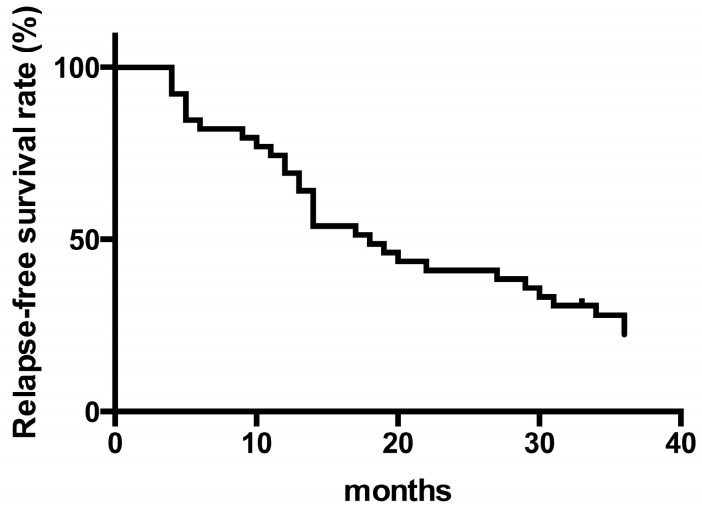
Kaplan–Meier plots for relapse-free survival of patients treated with tacrolimus for more than 3 months (n = 39).

**Figure 3 jcm-09-01771-f003:**
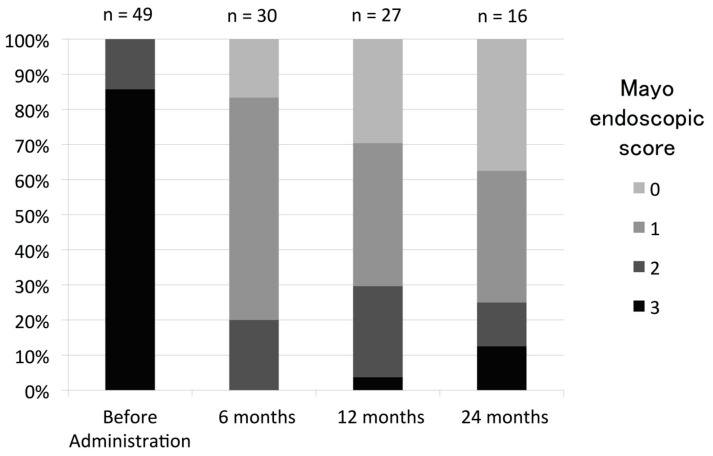
Endoscopic evaluation of patients treated with tacrolimus. Endoscopic severity was assessed by endoscopic-activity index before tacrolimus therapy, and 6, 12, and 24 months after administration.

**Figure 4 jcm-09-01771-f004:**
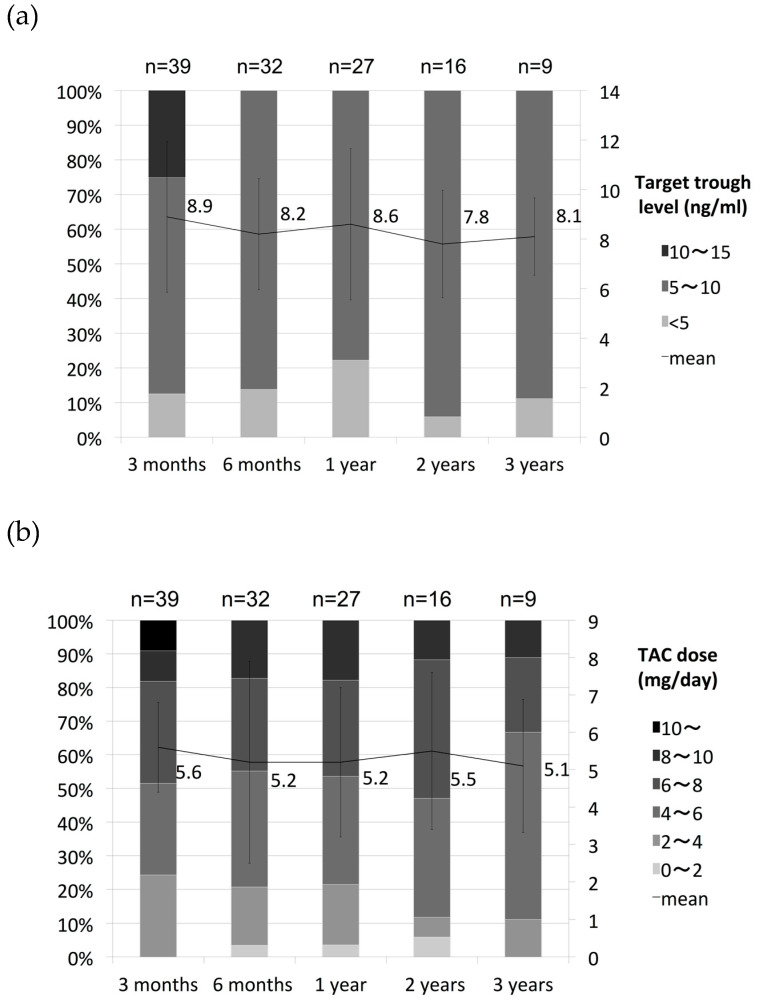
Transition of tacrolimus (TAC) trough level and dose. (**a**) Target through level; (**b**) TAC does. Whole-blood levels of TAC were maintained at a range of 5–10 ng/mL after trough level of 10–15 ng/mL for first 2 weeks to induce remission.

**Figure 5 jcm-09-01771-f005:**
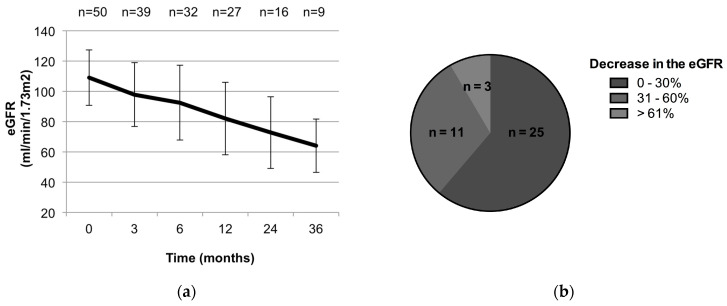
Changes in renal function. (**a**) Chronological changes in estimated glomerular filtration rate (eGFR) after administration of tacrolimus. (**b**) Decrease in eGFR during observation period (n = 39).

**Figure 6 jcm-09-01771-f006:**
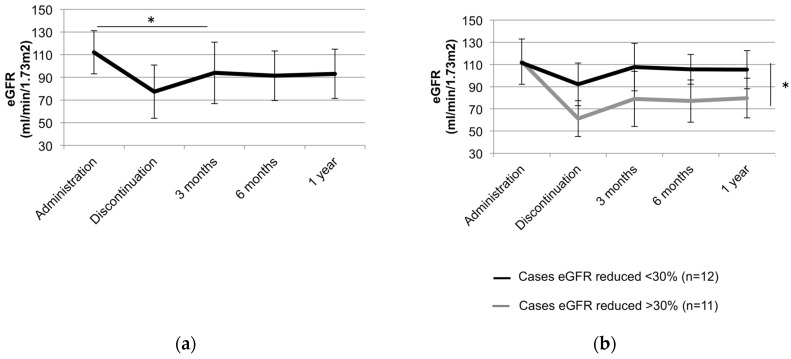
Changes in estimated glomerular filtration rate (eGFR) after tacrolimus (TAC) discontinuation. (**a**) Changes in eGFR in overall patient population. Changes in renal function were analyzed in 23 patients treated with TAC for more than 6 months and whose course could be followed for more than 1 year after discontinuation of TAC treatment. * *p* < 0.05 (analyzed using Mann–Whitney test). (**b**) Changes in eGFR in patients with eGFR reductions less than 30% (n = 12) and in patients with eGFR reductions more than 30% (n = 11) after administration. * *p* < 0.05 (analyzed using Mann–Whitney’s test).

**Table 1 jcm-09-01771-t001:** Patient characteristics.

**Age (Median (Range)) (year)**	37.5 (18–68)
**Gender**	
Male, n (%)	29 (58%)
Female, n (%)	21 (42%)
**Disease duration (median (range)) (year)**	6 (1–33)
**Location of colitis**	
Left-sided colitis, n (%)	19 (38%)
Extensive colitis, n (%)	31 (62%)
**Response to corticosteroid**	
Steroid-refractory, n (%)	14 (28%)
Steroid-dependent, n (%)	34 (68%)
Steroid-Naïve, n (%)	2 (4%)
**Clinical activity (median (range))**	
Lichtiger clinical-activity index	12 (8–16)
**Data at start of treatment**	
Hemoglobin (g/dL)	11.4 ± 2.3
C-reactive protein (mg/dL)	2.6 ± 4.1
eGFR ^1^ (mL/min/1.73 m^2^)	108.1 ± 25.2
Serum creatinine (mg/dL)	0.6 ± 0.1
**Duration of TAC ^2^ treatment (median (range)) (months)**	13.5 (1–64)
**Medications prior to TAC, n**	
5-Aminosalicylate	45
Corticosteroids	36
Thiopurine	24
Cytapheresis	19
Anti-TNF ^3^	4

^1^ eGFR, estimated glomerular filtration rate; ^2^ TAC, tacrolimus; ^3^ TNF, tumor necrosis factor.

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
