# Peer review of "Effectiveness and Nephrotoxicity of Long-Term Tacrolimus Administration in Patients with Ulcerative Colitis"

_jcm, 2020, doi:10.3390/jcm9061771_

Round 1

Reviewer 1 Report

General: The authors have identified an interesting research question. “Effectiveness and nephrotoxicity of long-term tacrolimus administration in patients with ulcerative colitis” is an interesting topic. Changes which are required are as below.

Title:

Looks ok

Abstract:

Please change the sentence “Regrettably, the complex mechanisms of LD disappearance have not been fully elucidated.” by not using the word regrettably.

Please expand all the short forms throughout the manuscript or when they are used first

Introduction:

Mechanism of action of tacrolimus in helping Ulcerative colitis will be helpful

Materials and Methods:

How was ulcerative colitis patients classified as mild,moderate and severe, this will need to be mentioned

Results:

Look ok

Discussion Section:

Authors need to discuss in detail about the limitations of the study.

Please include this important study https://pubmed.ncbi.nlm.nih.gov/28617029/ in discussion

Last paragraph can be made into a separate conclusion section.

Tables are appropriate.

English and grammar need to be thoroughly checked

Overall a well conducted study.

Author Response

Point-by-point reply to reviewers’ comments. (jcm-826367)

We thank the reviewers for their constructive and insightful comments for improving our manuscript. Below we describe a point by point reply and indicate the changes in the text to address these points. These changes improve the manuscript which we hope might now be acceptable for publication. 

Reviewer: 1

Point #1. Changethe sentence “Regrettably, the complex mechanisms of LD disappearance have not been fully elucidated.” by not using the word regrettably.

Response: We have searched carefully, but we are afraid that the sentence “Regrettably, the complex mechanisms of LD disappearance have not been fully elucidated.” is not in our abstract.

Point #2.Please expand all the short forms throughout the manuscript or when they are used first.

Response: As the reviewer pointed out, we have added "tumor necrosis factor -α" when we first used "TNF-α" in the main text and table. (page 1 line35, and Table1)

Point #3.Mechanism of action of tacrolimus in helping Ulcerative colitis will be helpful.

Response: TAC has immunosuppressive effects by suppressing cellular functions such as cytokine productionthrough T cell activation, and it is well known that T cell dysfunction plays a crucial role in the pathogenesis of IBD. We have added the mechanism of action of tacrolimus, and made changes in the text.(page 1 line 36 thru line 39)

Point #4.How was ulcerative colitis patients classified as mild, moderate and severe , this will need to be mentioned.

Response: As the reviewer suggested, we have added the information on severity classification; CAI >10, 7-9, and CAI <6 were defined as severe, moderate and mild respectively.We added reference #26 including this information. (page 2 line 80 thru line 81)

Point #5. Authors need to discuss in detail about the limitations of the study.

Response: As the reviewer suggested, we have described additional notes on thelimitations of our study and made changes in the text; limitations of our study include the sample size, retrospective design, single-center experience, and heterogeneity of previous and current treatment among patients. Also, urine tests were not examined in this study, and further studies are needed to elucidate the cause of irreversible renal damage due to long-term administration of TAC. (page 7 line 223 thru line 226)

Point #6.Please include this important study in discussion.

https://pubmed.ncbi.nlm.nih.gov/28617029/

Response: As the reviewer suggested,we have addedreference #20 in the manuscript, and included this study in discussion.(page 2 line 45 thru line 46, and page 7 line 183 thru line 185)

Point #7. Last paragraph can be made into a separate conclusion section.

Response: As the reviewer suggested,we have moved the following sentence into the conclusion section. “There are still no clinical trials that prospectively examined the effectiveness of long-term administration of TAC, and it is necessary to accumulate evidence on the usefulness and safety of TAC dosing for the maintenance of remission.” (page 7 line 227 thru page 8 line 230)

Point #8. English and grammar need to be thoroughly checked.

Response: Our manuscript has been reviewed by an experienced medical editor whose first language is English. Attached please find the proof of review.

Reviewer 2 Report

This manuscript aims to study the effects of Tacrolimus on renal function for UC patients. To answer this question, the authors investigated 50 UC patients with TAC treatment. Clinical response, eGFR, and serum creatinine levels were tested. The authors found that TAC is an effective option for ameliorating UC disease while TAC increases the risk of nephrotoxicity. Although this study is an interesting research topic and organized in good shape, the manuscript does need minor work before publication.

  1. How about the history of kidney injury/diseases of UC patients?
  2. How about the family history of inflammatory bowel disease and renal diseases among UC patients.

Author Response

Point-by-point reply to reviewers’ comments. (jcm-826367)

We thank the reviewers for their constructive and insightful comments for improving our manuscript. Below we describe a point by point reply and indicate the changes in the text to address these points. These changes improve the manuscript which we hope might now be acceptable for publication. 

Reviewer: 2

Point #1. How about the history of kidney injury/diseases of UC patients?

Point #2. How about the family history of inflammatory bowel disease and renal diseases among UC patients.

Response: We have added the following sentence including the information about medical history or family history of renal disease and IBD. “No patient had a history of renal disease. Five patients had family history of inflammatory bowel disease, and one patient of renal disease. ” (page 3 line 111 thru line 112)

Reviewer 3 Report

The paper evaluated the long-term efficacy and safety of tacrolimus for maintaining remission in ulcerative colitis patients. Indeed, short-term efficacy and safety has been investigated but reports on long-term use are still few.

The Materials and methods section is appropriate, Ethics Statement is clearly detailed. Statistics are properly applied.

The results description is well organized, and figures and tables are carefully elaborated.

English language is fine.

Author Response

Point-by-point reply to reviewers’ comments. (jcm-826367)

We thank the reviewers for their constructive and insightful comments for improving our manuscript. Below we describe a point by point reply and indicate the changes in the text to address these points. These changes improve the manuscript which we hope might now be acceptable for publication.
